# Three-Dimensional Lattice Structure to Reduce Parasitic Inductance for WBG Power Semiconductor-Based Converters

Sung-Soo Min, Chan-Hyeok Eom, Yeong-Seop Jang and Rae-Young Kim *

Department of Electrical and Biomedical Engineering, Hanyang University, Seoul 04763, Republic of Korea; minss7160@hanyang.ac.kr (S.-S.M.); chan3403@hanyang.ac.kr (C.-H.E.); jys0531@hanyang.ac.kr (Y.-S.J.)
* Correspondence: rykim@hanyang.ac.kr; Tel.: +82-2-2220-2897

**Abstract:** Wide bandgap (WBG) power semiconductors can achieve high efficiency and power density due to their low on-resistance and fast switching speeds. However, the fast-switching speed induces voltage to the parasitic inductance in the circuit, causing a significant overshoot in the drain-source voltage of the devices and the ringing of the drain current due to resonance with the parasitic capacitance. Thus, minimizing parasitic inductance is necessary for driving WBG power semiconductors in a stable manner. This paper proposes a three-dimensional lattice structure that reduces parasitic inductance through horizontal and vertical magnetic flux cancellations within a printed circuit board (PCB). The relationship between the magnetic flux cancellation and the parasitic inductance is analyzed, and the magnetic flux cancellation in the proposed structure is described. In addition, a practical PCB layout design procedure based on the proposed structure is provided. Simulation results demonstrate a 55.8% reduction in parasitic inductance, and experimental results show reduced overshoot and ringing at the switching transient, resulting in a 26% reduction in switching loss. As a result, the proposed method can improve the efficiency and stability of WBG device-based power converters.

**Keywords:** magnetic flux cancellation; parasitic inductance; PCB layout; wide bandgap power semiconductor

## 1. Introduction

Power semiconductor devices account for the largest portion of power loss in power converters [1]. Conventional silicon (Si) power semiconductors are limited by breakdown voltage, operating temperature, and switching frequency, which significantly reduce the efficiency of power converters and necessitate complex and expensive cooling systems. Recently, wide bandgap (WBG) semiconductors, which include silicon carbide (SiC) and gallium nitride (GaN), with excellent characteristics, such as a large critical electric field, high electron mobility, and high thermal conductivity have been developed to overcome existing physical limitations. The superior characteristics of WBG semiconductors include low on-resistance, high switching speed, and high operating temperature to power semiconductor devices. Thus, power converters can achieve high efficiency and power density [1–5].

However, to exploit WBG power semiconductors, more factors must be considered when using Si devices. WBG power semiconductors have a characteristic of fast switching speed attributed to their small input and output capacitances, while the disadvantages of a large $di/dt$ and $dv/dt$. Therefore, WBG power semiconductors are significantly affected by parasitic inductance during switching transients, and this effect appears as ringing, undershooting, and overshooting of the voltage and current. These effects may increase the switching loss and weaken the advantage of WBG power semiconductors or even cause failure [6–9].

Therefore, minimizing the parasitic inductance of the printed circuit board (PCB) is necessary to operate WBG devices in a stable manner for increasing the efficiency and power density of the power converter by utilizing WBG power semiconductors.

Thus far, various methods have been proposed to minimize parasitic inductance. For example, it can be reduced by increasing the width of the conductor through which the current flows, or by reducing the length of the current loop [10–15]. The length of the power loop is minimized by directly mounting the decoupling capacitor onto a system-in-package (SiP) that comprises a half-bridge leg [10]. This method can reduce the power loop inductance; however, soldering the capacitor directly to a SiP is complicated. In [11], the capacitor was mounted directly on the die of a switching device, similar to that in [10]. In [12,13], the current loop was significantly reduced by embedding components and devices into the PCB. Nielsen et al. [12] proposed a structure that involves inserting capacitors vertically into the PCB, adjacent to the GaN-based half-bridge structure. Qi et al. [13] suggested a half-bridge module, in which a GaN bare die was embedded into the inner layer of the PCB. Both of these methods minimize parasitic inductance by considerably reducing the current loop. However, they are complicated to implement because a specialized process is required. Lu et al. [14] proposed a half-bridge module consisting of a decoupling capacitor and two GaN devices. Unlike methods proposed in [12,13], the proposed method is relatively simple to implement because there is no need for an additional process in the PCB. The method also maintains low inductance due to the placement of the decoupling capacitor in close proximity. However, it is only applicable to half-bridge-based converters and cannot be used for multi-level converters, such as Neutral Point Clamped inverters. The above methods are difficult to apply in the design of general power converters because they can only be applied to special packages or applications.

Accordingly, parasitic inductance reduction methods that use magnetic flux cancellation between adjacent conductors, which can be applied in the general case, have been studied [15–21]. In [15–17], structures can reduce parasitic inductance through single current loop-based magnetic flux cancellation. Reusch et al. [15] and Reusch and Strydom [16] placed conductors with opposite current directions close to each other to improve the magnetic flux cancellation. In [17], the authors designed prototypes based on three types of single loop-based inductance reduction methods and compared their performance. The proposed methods successfully reduced parasitic inductance through improved magnetic flux cancellation; however, these methods have limitations in reducing parasitic inductance because they consider only a single current loop formed between the top and bottom layers, or in a single layer of the PCB.

In [18–21], multiple current loop-based inductance reduction methods were proposed to improve flux cancellation. In [18], a switching package capable of constructing multiple current loops in a single layer was proposed. The package is configured to form several branches with currents in different directions adjacent to each other, allowing for reduced parasitic inductance. However, this is only applicable to certain package types of devices that can form multiple current loops. In [19–21], more general multiple loop-based methods were proposed. Dechant et al. [19] and Hammer et al. [20] proposed structures that can improve the magnetic flux cancellation in multiple current loops without the limitation of package type. The proposed method improved magnetic flux cancellation between conductors by reversing the direction of the current flowing in adjacent layers in a multilayer PCB. However, these articles only presented a conceptual method and did not provide a quantitative interpretation of the improvement of inductance reduction by a multi-loop structure, nor did they provide a detailed design process based on it. Yang et al. [21] completed the foundation of a multi-loop-based inductance reduction method by providing a detailed principle of inductance reduction by a multi-loop structure and its design method. However, the above methods [19–21] did not fully utilize the inductance reduction effect of multi-loop structures because they only considered magnetic flux offset between layers, without considering the layers themselves.

In this study, a three-dimensional (3-D) lattice structure that can cancel the magnetic flux between the layers and in a single layer is proposed to further reduce parasitic inductance. The basic principle of magnetic flux cancellation is described, and a 3-D lattice structure is presented based on this principle. The proposed structure involves arranging a pair of adjacent PCB layers so that their current directions are opposite, leading to the cancellation of the vertical magnetic flux. Additionally, segments are inserted into each layer to generate horizontal flux cancellation. Subsequently, a pair of layers that can cancel both the vertical and horizontal fluxes are designated as reference layers and are repeated to form a multi-loop, which further enhances magnetic flux cancellation. The inductance reduction performance of the 3-D lattice structure is verified using ANSYS Q3D, and a detailed PCB layout design based on the proposed structure is also presented. Finally, improved switching performances, such as ringing, overshoot, and switching energies are experimentally verified.

## 2. Proposed Parasitic Inductance Reduction Structure with Three-Dimensional Magnetic Flux Cancellation

### 2.1. Relationship between Parasitic Inductance and Magnetic Flux Cancellation

Figure 1a shows the magnetic flux generated when two conductors with the same current direction on the PCB are adjacent. The green plane represents FR-4 and the yellow plane represents the conductors. The two conductors have a length $l$ and are spaced apart from each other by the distance $d$ with currents $i_a$ and $i_b$ flowing in the same direction. The blue dotted line represents the magnetic flux generated by $i_a$ and the red dotted line represents the magnetic flux generated by $i_b$. Figure 1a shows that the magnetic fluxes are canceled by each other between the conductors and are added outside the conductors when two conductors with the same current direction are adjacent. Therefore, the two conductors must be far apart to increase the area in which magnetic fluxes are canceled to reduce parasitic inductance. Figure 1b shows the electrical equivalent model of Figure 1a. $L_a$ and $L_b$ represent the self-inductances of each conductor, and $M_{ab}$ represents the mutual inductance between the two conductors. The induced voltage $v_a$ and $v_b$ in each conductor by $i_a$ and $i_b$ can be expressed by Faraday's law as:

$$\begin{cases} v_a = L_a \frac{di_a}{dt} + M_{ab} \frac{di_b}{dt} \\ v_b = L_b \frac{di_b}{dt} + M_{ab} \frac{di_a}{dt} \end{cases} \tag{1}$$

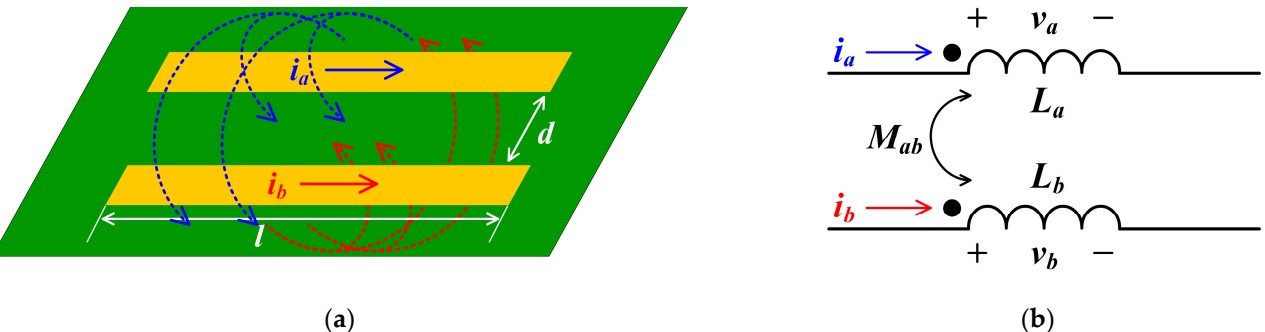

(**a**)           (**b**)

**Figure 1.** Magnetic flux and inductances of two conductors with the same current direction. (**a**) Current and magnetic flux flow; (**b**) equivalent circuit.

If the magnitudes of $i_a$ and $i_b$ are equal, the induced voltages can be expressed as:

$$\begin{cases} v_a = (L_a + M_{ab}) \frac{di_a}{dt} \\ v_b = (L_b + M_{ab}) \frac{di_b}{dt} \end{cases} \tag{2}$$

Therefore, when two conductors with the same current direction are adjacent, the effective inductance of each conductor is expressed as:

$$\begin{cases} L_{a,e} = L_a + M_{ab} \\ L_{b,e} = L_b + M_{ab} \end{cases} \tag{3}$$

where $L_{a,e}$ and $L_{b,e}$ denote the effective inductances of the conductors.

According to (2) and (3), mutual inductance must be minimized because it increases the effective inductance. If $l$ is sufficiently longer than $d$, the mutual inductance $M$ can be expressed as:

$$M \simeq \frac{\mu_0}{2\pi} \cdot l \cdot \left( \ln \frac{2l}{d} - 1 \right) \tag{4}$$

where $\mu_0$ represents the magnetic permeability of free space [21,22].

According to (4), the mutual inductance decreases with a decrease in the length of the conductor and an increase in the distance between the conductors. In other words, the conductors with the same current direction must be separated to reduce parasitic inductance.

Figure 2a shows the magnetic flux generated when two conductors with opposite current directions are adjacent. Unlike in Figure 1a, magnetic fluxes are added between the conductors and canceled outside. Therefore, the distance between two conductors must be reduced. Figure 2b shows the electrical equivalent model of Figure 2a. The induced voltage $v_c$ and $v_d$ in each conductor by currents $i_c$ and $i_d$ can be expressed using the same principles as (1) and (2):

$$\begin{cases} v_c = (L_c - M_{cd}) \frac{di_c}{dt} \\ v_d = (L_d - M_{cd}) \frac{di_d}{dt} \end{cases} \tag{5}$$

where $L_c$ and $L_d$ represent the self-inductance of each conductor and $M_{cd}$ represents the mutual inductance between the two conductors.

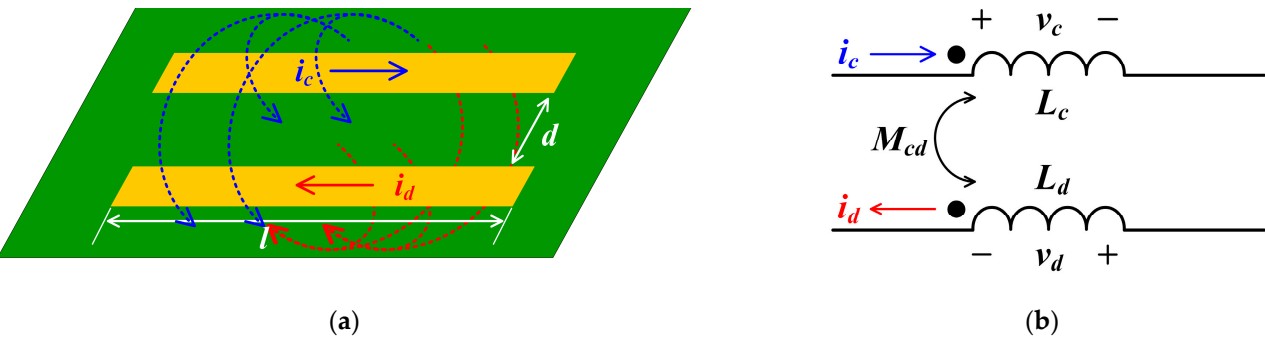

(**a**)                                    (**b**)

**Figure 2.** Magnetic flux and inductances of two conductors with opposite current directions. (**a**) Current and magnetic flux flow; (**b**) equivalent circuit.

Therefore, when two conductors with opposite current directions are adjacent, the effective inductance of each conductor is expressed as:

$$\begin{cases} L_{c,e} = L_c - M_{cd} \\ L_{d,e} = L_d - M_{cd} \end{cases} \tag{6}$$

where $L_{c,e}$ and $L_{d,e}$ denote the effective inductances of the conductors.

Unlike (3), the larger the mutual inductance, the smaller the effective inductance when two conductors with opposite current directions are adjacent. Therefore, parasitic inductance can be reduced further by placing the two conductors close to each other.

In summary, to effectively reduce parasitic inductance, conductors of the same current direction should be placed farther apart to reduce the mutual inductance, and conductors

in opposite current directions should be placed closer to each other to increase the mutual inductance.

### 2.2. Proposed Three-Dimensional Lattice Structure

Figure 3a shows the conceptual configuration of the proposed 3-D lattice structure. The blue and red blocks represent the conductor segments, and the yellow and white arrows represent the directions of the current in each segment, where $s_{nm}$ represents the mth segment of the nth layer. In the first layer, conductors with opposite current directions are placed adjacent to each other to improve the magnetic flux cancellation within the layer. In the second layer, the segments are arranged, such that the current direction is opposite to that of the vertically adjacent layer, to improve interlayer flux cancellation. Subsequently, the first and second layers are set in pairs and stacked repeatedly to form a multilayer structure to enhance the magnetic flux cancellation. Figure 3b shows the magnetic flux cancellation between adjacent layers in the proposed structure. Horizontal magnetic flux cancellation occurs between two adjacent segments within a layer, and vertical flux cancellation occurs between adjacent layers. Therefore, the proposed 3-D lattice structure can effectively reduce the parasitic inductance of the PCB by generating both vertical and horizontal magnetic flux cancellations, which can increase with an increase in the number of segments or layers.

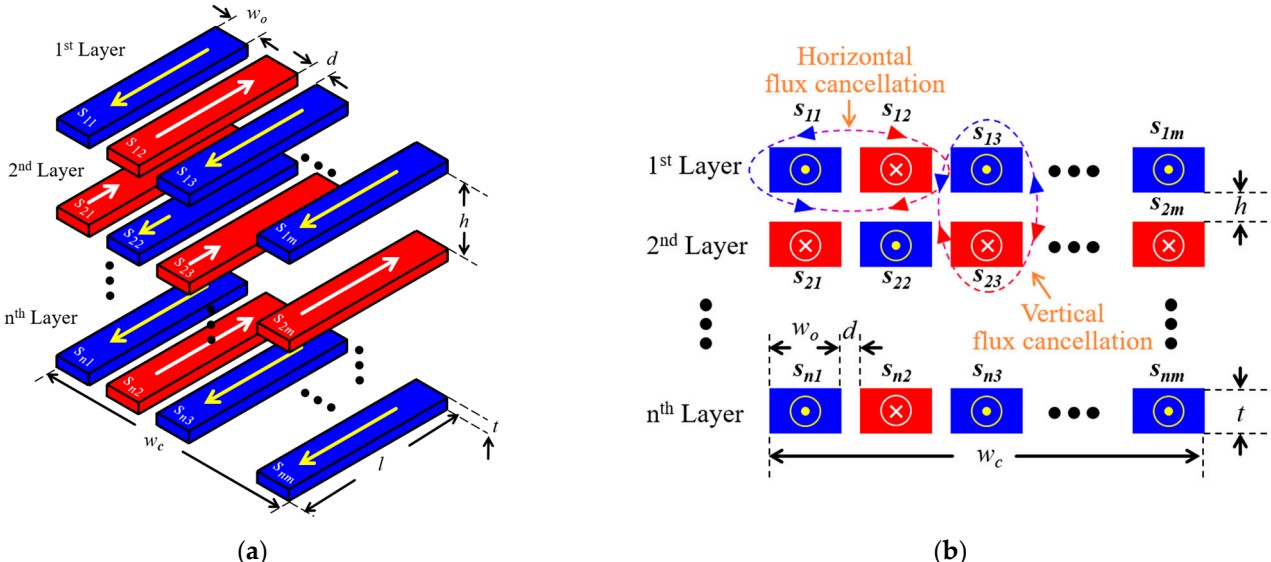

**Figure 3.** Proposed three-dimensional lattice structure. (**a**) Conceptual configuration; (**b**) magnetic flux cancellation in the proposed structure.

Figure 4 shows a comparison of the magnetic flux density in the conductor before and after the application of the proposed structure using ANSYS MAXWELL. In both cases, the width $w_c$, thickness $t$, length $l$, and distance between layers $h$ is 54 mm, 10 mm, 200 mm, and 50 mm, respectively. The white arrow indicates the direction of the current flowing through the conductor. In Figure 4a, two conductors with opposite current directions are arranged vertically, which results in only vertical magnetic flux cancellation. As shown in Figure 4b, the proposed structure is applied by separating the conductor into five segments. The width of segment $w_o$ is 10 mm, and the distance $d$ between the segments is 1 mm. The flux density of the conductor is reduced significantly, compared to that in Figure 4a, because of the horizontal and vertical flux cancellations. This implies that the proposed structure has a smaller parasitic inductance. Table 1 lists the simulated parasitic inductances of the two cases shown in Figure 4 using ANSYS Q3D. As summarized in Table 1, the parasitic inductance of the proposed structure is reduced by 55.8% to 43.38 nH compared to 98.2 nH, when only magnetic flux is canceled vertically.

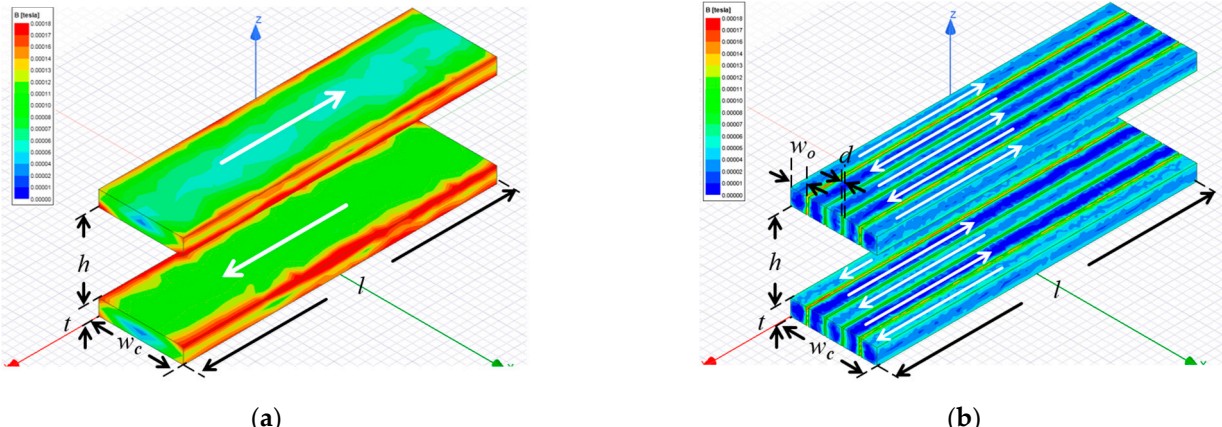

**Figure 4.** Simulated magnetic flux density of conductors. (**a**) With only vertical flux cancellations; (**b**) with both vertical and horizontal flux cancellations.

**Table 1.** Comparison of parasitic inductance.

|  | w/o Proposed Structure (Figure 4a) | w/Proposed Structure (Figure 4b) |
|---|---|---|
| Simulated inductance | 98.2 nH | 43.38 nH |

Figure 5 shows the design procedure for the proposed 3-D lattice structure. White and yellow arrows indicate the current direction. The proposed structure is designed as follows:

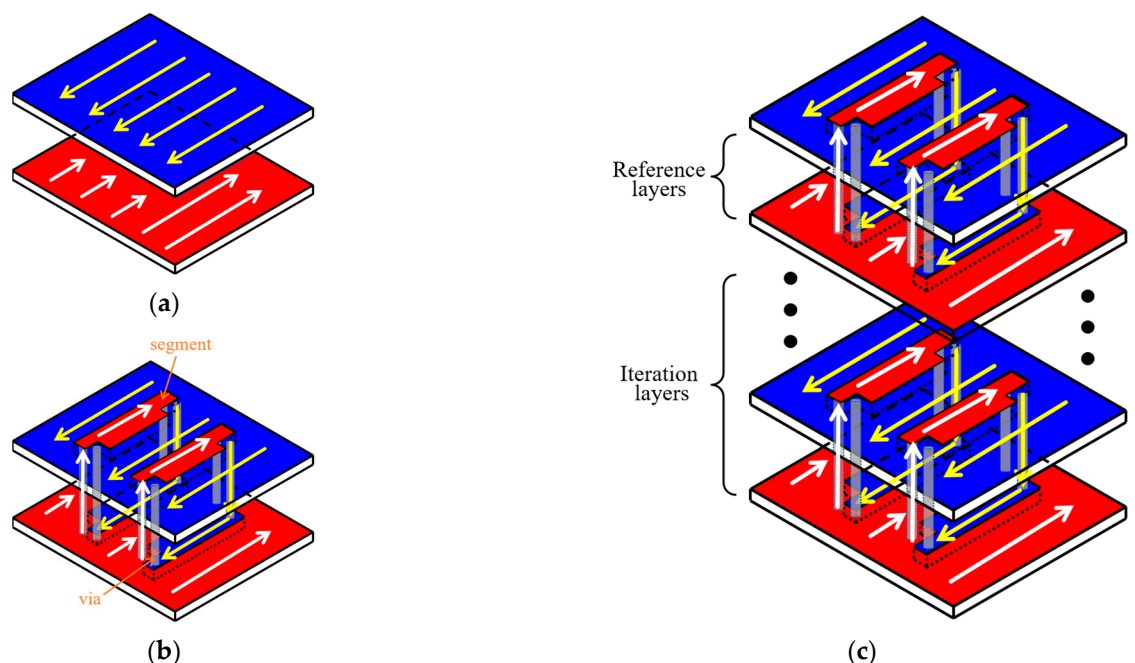

**Figure 5.** Design procedure of the proposed structure. (**a**) A pair of layers with vertical flux cancellation; (**b**) a pair of layers with three-dimensional flux cancellation; (**c**) proposed lattice structure.

First, a pair of layers is formed, in which the current flows in opposite directions to enable vertical magnetic flux cancellation. At this stage, it is important to design a PCB that maximizes the area in which the 3-D lattice structure can be applied, and this area depends on the arrangement of switching devices and decoupling capacitors. If the switching devices and decoupling capacitors are located on the same layer, such as in the

lateral structure in [12], it is difficult to apply the proposed structure because it forms a closed current loop within the layer. In contrast, if the switching devices and capacitors are located on different layers (e.g., top layer: devices and bottom layer: capacitors), it is easy to form a pair of layers, as shown in Figure 5a, and more segments can be formed to reduce parasitic inductance.

Second, segments are formed, such that conductors with opposite current directions are adjacent within the layer. As shown in Figure 5b, the two layers are connected to allow current to flow through the segments. The current in the lower layer flows through the via to the segments in the upper layer; as a result, three and two current paths are formed in the left and right directions alternately in the upper layer, respectively. Similar to the upper layer, three current paths in the right direction and two current paths in the left direction are formed in the lower layer. Through this structure, the parasitic inductance can be reduced by generating not only vertical flux cancellation between layers but also horizontal flux cancellation within a layer.

When the location of where to form the segments is determined, $w_o$ and $d$ can be determined. If the distance between the segments is too small, the PCB may suffer from insulation problems. Therefore, $d$ should be determined by considering standards such as IPC-2221 or IPC-9592. According to the IPC-2221 standard, $d$ must satisfy:

$$\begin{cases} d > 0.25 + 0.0025 \cdot \left(V_{pk} - 500\right) & \text{(internal layer)} \\ d > 0.8 + 0.00306 \cdot \left(V_{pk} - 500\right) & \text{(external layer)} \end{cases} \tag{7}$$

where $V_{pk}$ represents the maximum voltage applied between the segments and the unit of $d$ is mm.

Meanwhile, with an increase in the number of segments, the area in which the current can flow in the conductor decreases because of the clearance for insulation. Therefore, the number of segments $m$ in a layer should be determined by considering the temperature increase in the conductor of the PCB. According to the IPC-2221 standard, $m$ is determined as:

$$m < \frac{1}{d} \left\{ w_c - \left( \frac{I_x}{k \cdot \Delta T^b} \right)^{1/c} / t \right\} \tag{8}$$

where $I_x$, $\Delta T$, and $k$, $b$, and $c$ represent the magnitude of the current, allowable temperature rise, and coefficients of layer, respectively.

Third, a pair of layers in Figure 5b is set as the reference layer, as shown in Figure 5c, and the reference layer is repeated as many times as desired. Parasitic inductance reduces with an increase in the number of iteration layers. However, this increases the cost of the PCB design, and therefore, the number of iteration layers should be determined by considering this point.

## 3. Detailed PCB Layout Design of Proposed Three-Dimensional Lattice Structure

Figure 6 shows the parasitic inductance and current flow of a half-bridge leg during switching transients. The parasitic inductance consists of the high-side drain inductance $L_{d,H}$ of high-side devices $Q_1$, high-side source inductance $L_{s,H}$, low-side drain inductance $L_{d,L}$ of low-side devices $Q_2$, and low-side source inductance $L_{s,L}$. The current during the switching transients includes current $I_{C2Q}$ flowing from the decoupling capacitor $C_{dec}$ to $Q_1$, a current $I_{Q2Q}$ flowing from $Q_1$ to $Q_2$, and a current $I_{Q2C}$ flowing from $Q_2$ to $C_{dec}$. The direction of the current can be reversed based on the switching state of the device.

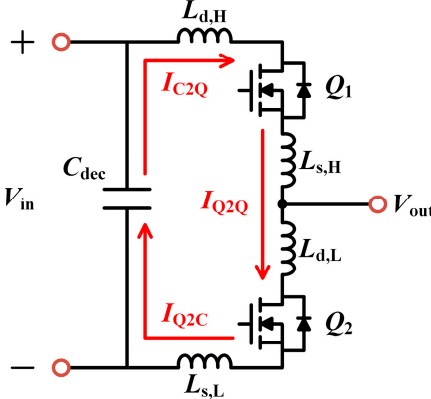

**Figure 6.** Parasitic inductance and current flow of a half-bridge leg during switching transients.

Figure 7 shows the four-layer PCB layout design with the proposed structure, which considers the parasitic inductance and current flow of the half-bridge leg. The design specifications are listed in Table 2. $V_{pk}$ and $I_x$ were set to 650 V and 30 A, respectively, because the 650 V/30 A GaN power semiconductor GS66508T was selected as the switching device. Further, $w_c$, $t$, and $\Delta T$ were set as 18 mm, 0.07 mm, and 15 °C, respectively. While $d$ was set as 0.625 mm based on (7), and the proposed structure was applied to only the inner layer because the external layer required a considerably wider clearance than the inner layer. Finally, $m$ was selected as 5 based on (8), and $w_o$ was selected as 3.1 mm so that the 5 segments with 0.625 mm of $d$ could be applied to an 18 mm conductor.

**Table 2.** Design specifications.

| Parameters | Value |
| --- | --- |
| $V_{pk}$ | 650 V |
| $I_x$ | 30 A |
| $w_c$ | 18 mm |
| $t$ | 0.07 mm |
| $\Delta T$ | 15 °C |
| $d$ | 0.625 mm |
| $m$ | 5 |
| $w_o$ | 3.1 mm |

In the top layer, $C_{dec}$ and the input voltage $V_{in}$ plane exist, and $I_{C2Q}$ and $I_{Q2C}$ flow upward, as shown in Figure 7a. Figure 7b shows the PCB layout and current flow of inner layer 1. In inner layer 1, the output voltage $V_{out}$ plane exists, and $I_{Q2Q}$ flows downward during the switching transients. Therefore, magnetic fluxes generated in the top and inner layers 1 cancel each other. Furthermore, $I_{C2Q}$ and $I_{Q2C}$ in the top and inner layer 2 flow through the vias into the segments, which results in horizontal magnetic flux cancellation. In inner layer 2, the $V_{in}$ plane exists and $I_{C2Q}$ and $I_{Q2C}$ flow upward, similar to the top layer, which results in vertical flux cancellation with inner layer 1, as shown in Figure 7c.

Segments were also added to generate horizontal flux cancellation, as in Figure 7b. Finally, $Q_1$ and $Q_2$ were mounted on the bottom layer, and $I_{Q2Q}$ flowed to generate a vertical flux cancellation with inner layer 1. As explained in the design procedure, the parasitic inductance can be reduced further if inner layers 1 and 2 are set as reference layers and stacked between the top and bottom layers.

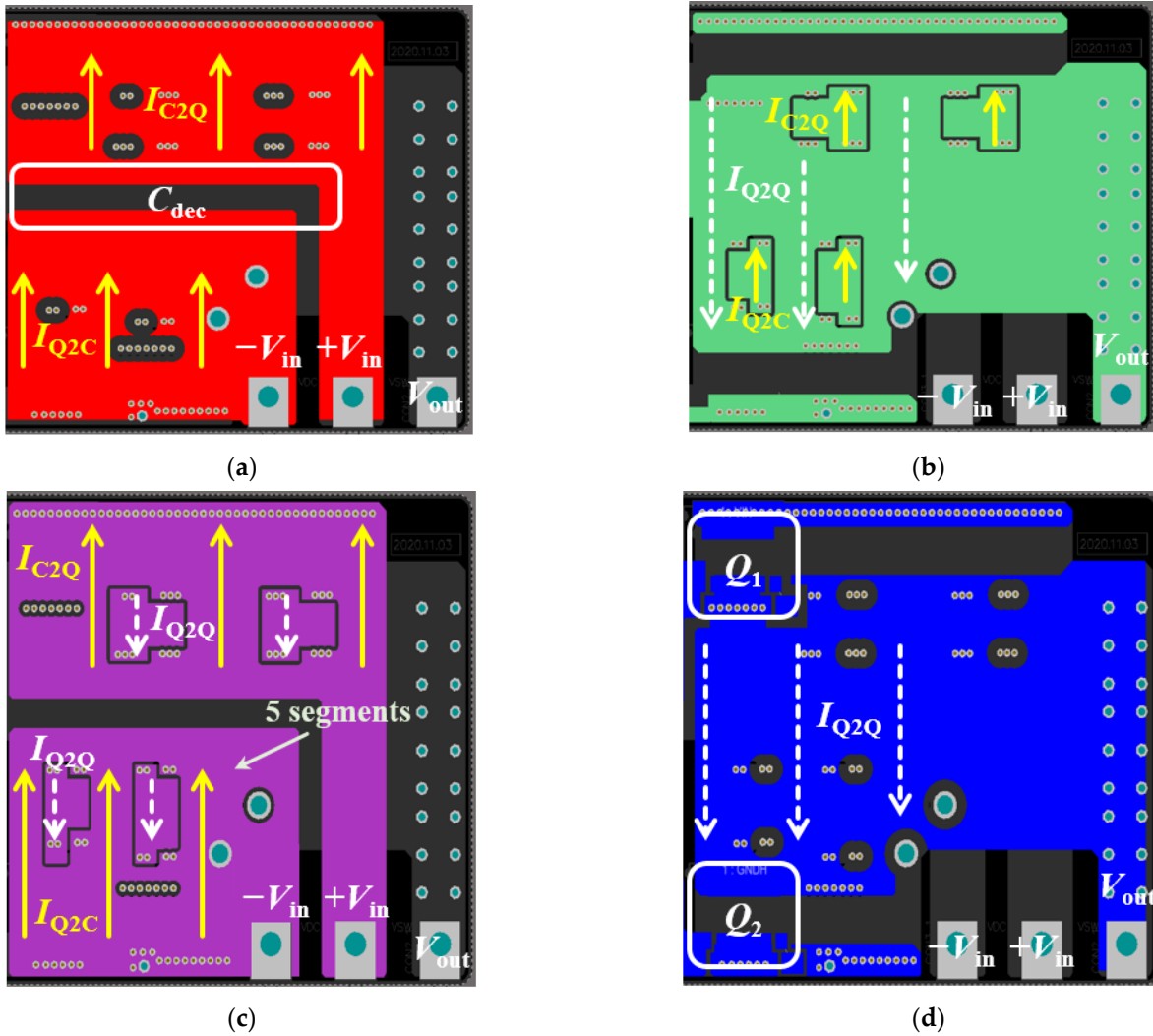

**Figure 7.** A four-layered PCB layout design with the proposed structure. (**a**) Top layer; (**b**) inner layer 1; (**c**) inner layer 2; (**d**) bottom layer.

Table 3 presents a comparison of the parasitic inductance between the conventional [21] and proposed structures. The conventional structure was applied to a four-layer PCB, as shown in the proposed structure in Figure 7, and only the vertical magnetic flux cancellation was considered. As summarized in Table 3, the total parasitic inductance $L_{tot}$ of the proposed structure was reduced by 24% to 8.26 nH compared to 10.85 nH.

**Table 3.** Comparison of parasitic inductance between the conventional structure in Yang et al. [21] and the proposed structure.

| Parameters | Conventional Structure in [21] | Proposed Structure |
| --- | --- | --- |
| $L_{d,H}$ | 2.13 nH | 1.69 nH |
| $L_{s,H}$ | 0.815 nH | 0.615 nH |
| $L_{d,L}$ | 2.445 nH | 1.845 nH |
| $L_{s,L}$ | 5.46 nH | 4.11 nH |
| $L_{tot}$ | 10.85 nH | 8.26 nH |

## 4. Performance Verification

Figure 8 shows the prototype designed to validate the performance of the proposed structure. $C_{\text{dec}}$ exists in the top layer, $Q_1$ and $Q_2$ exist in the bottom layer, and segments for the 3-D lattice structure exist in inner layers 1 and 2.

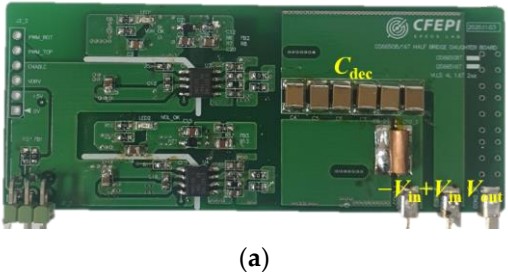

(**a**)

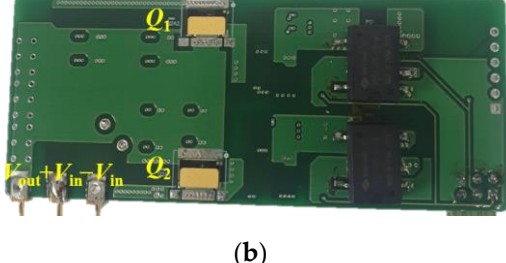

(**b**)

**Figure 8.** Photograph of the PCB with the proposed 3-D lattice structure. (**a**) Top view; (**b**) bottom view.

Figure 9 shows a circuit diagram and photograph of the experimental setup. A double-pulse test was performed to verify improvements in switching characteristics: the drain-source voltage $V_{\text{DS}}$ was measured using a LeCroy PP018 passive probe, and the drain current $I_{\text{D}}$ was measured with a Rogowski coil CWT-UM3/B/1/80. The system parameters are listed in Table 4.

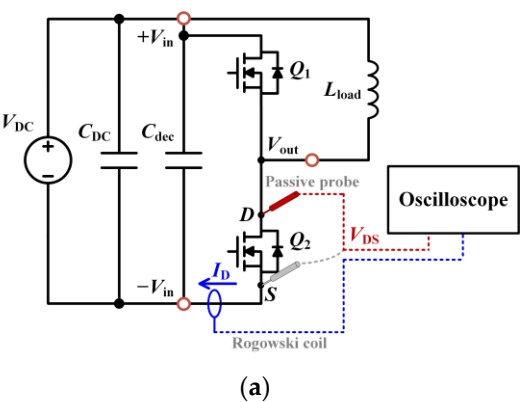

(**a**)

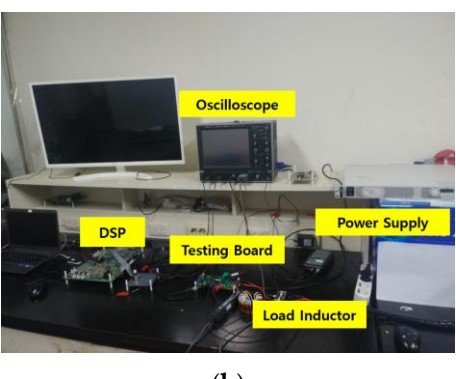

(**b**)

**Figure 9.** Experimental setup. (**a**) Circuit diagram; (**b**) photograph.

**Table 4.** System parameters.

| Parameters | Value |
| --- | --- |
| $V_{\text{DC}}$ | 300 V |
| $I_{\text{D}}$ | 21 A |
| $V_{\text{GS}}$ | 6/−3 V |
| $L_{\text{load}}$ | 200 µH |

Figure 10 shows the drain-source voltage and drain current of $Q_2$ during the turn-off transient. In the prototype with a conventional structure [21], $V_{\text{DS}}$ increases up to 380 V and $I_{\text{D}}$ shows a fluctuation range from 43 A to −20 A, as indicated in Figure 10a. Figure 10b shows the turn-off transient waveforms of the proposed structure. Compared with the conventional structure, the overshoot of $V_{\text{DS}}$ decreased to 359 V, and the maximum and minimum magnitude of $I_{\text{D}}$ decreased to 38 A and −13.5 A, respectively. The experimental results in Figure 10 indicate that the overshoot of $V_{\text{DS}}$ is reduced by approximately 5.5%, and the fluctuation of $I_{\text{D}}$ is reduced by approximately 18% with the proposed structure, meaning that the parasitic inductance is effectively reduced.

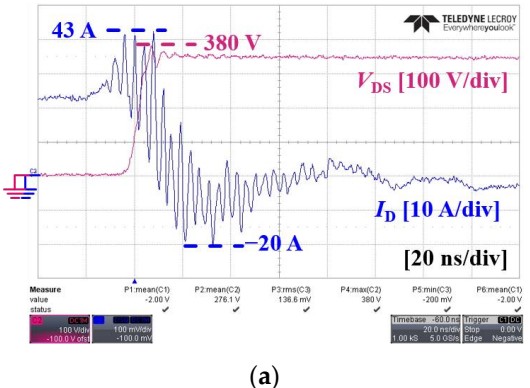

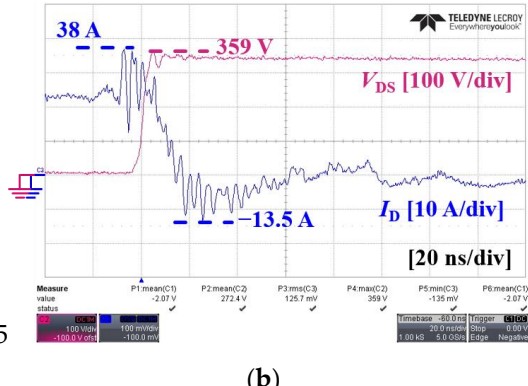

**(a)** **(b)**

**Figure 10.** Drain-source voltage and drain current waveform during the turn-off transient. (**a**) With the conventional structure [21]; (**b**) with the proposed structure.

Figure 11 shows the drain-source voltage and drain current of $Q_2$ during the turn-on transient. The maximum magnitude of $I_D$ decreased by 10% from 34 A to 31 A, and the minimum magnitude decreased by 35% from –10.6 A to –6.8 A with the proposed structure, as shown in Figure 11.

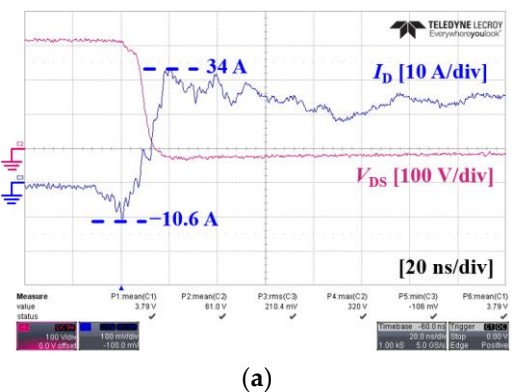

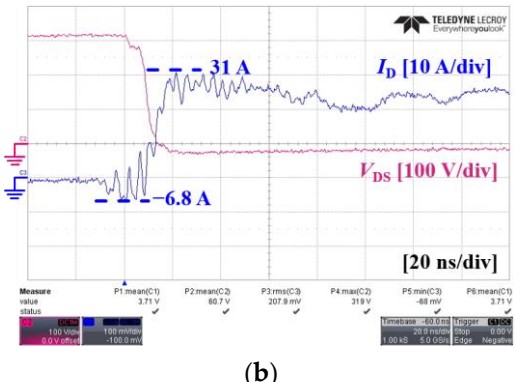

**(a)** **(b)**

**Figure 11.** Drain-source voltage and drain current waveform during the turn-on transient. (**a**) With the conventional structure [21]; (**b**) with the proposed structure.

Improving switching characteristics by reducing parasitic inductance can increase the efficiency of a power converter as it reduces the switching energy. Figure 12 shows a comparison of the switching energy for a conventional vertical lattice structure and that for the proposed structure, depending on the magnitude of $I_D$. The proposed structure has lower turn-on and turn-off switching energies than those of the conventional structure; the turn-on energy is reduced by approximately 10% and the turn-off energy by approximately 14%.

Figure 13 shows that the normalized switching energy depends on the inductance reduction method. The switching energy without inductance reduction was used as a reference for normalization. The proposed structure reduces the total switching energy by more than 10% compared to the conventional structure [21], and therefore, the switching energy is reduced by 26% compared to the case without an inductance reduction method.

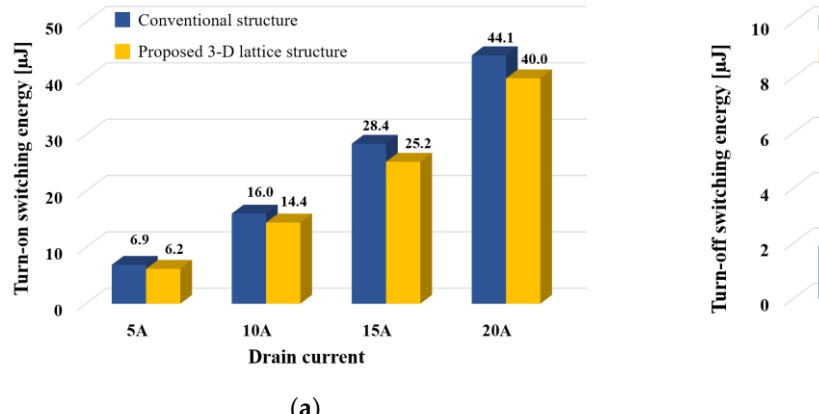
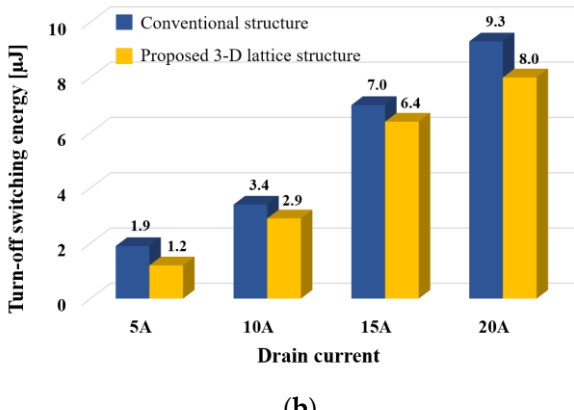

(**a**)  (**b**)

**Figure 12.** Comparison of switching energy for the conventional structure [21] and proposed structures. (**a**)Turn-on switching energy; (**b**) turn-off switching energy.

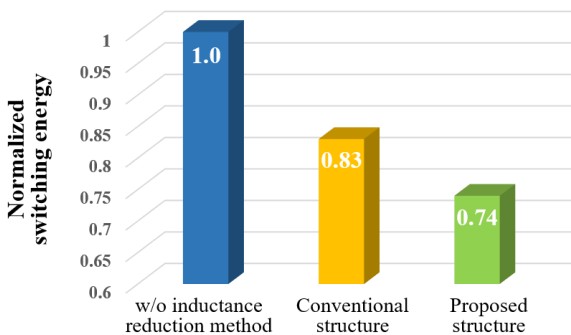

**Figure 13.** Normalized switching energy based on the inductance reduction method.

These experimental results confirm that the proposed structure can improve the stability and efficiency of WBG power semiconductors by reducing parasitic inductance without additional devices or control schemes.

## 5. Conclusions

A 3-D lattice structure was proposed to reduce parasitic inductance. The proposed 3-D lattice structure generated vertical magnetic flux cancellation by placing adjacent layers in opposite current directions, and the segments were inserted to generate the horizontal magnetic flux cancellation within a layer. A pair of layers, in which vertical and horizontal flux cancellations occur simultaneously, was selected as the reference layer and stacked repeatedly to maximize inductance reduction. Based on the simulation analysis, parasitic inductance was reduced by 55.8% with the proposed 3-D lattice structure. The design example demonstrated the selection of the segment parameters and application of the proposed structure to a practical PCB layout. The experimental results showed that the proposed structure improved switching characteristics, such as the overshoot and ringing of the drain-source voltage and drain current. The switching energy was reduced by 26% with the proposed structure compared to that without the inductance reduction method.

The proposed 3-D lattice structure demonstrates a significant reduction in parasitic inductance through a PCB layout design, making it applicable to WBG power semiconductor-based converters regardless of the package type of the switching devices. This allows for effective mitigation of the ringing and overshoots caused by the fast-switching speed of WBG power semiconductors, addressing the limitations of these devices and enabling high efficiency and power density operating with improved stability. However, it should be noted that the proposed structure may have limitations in higher power levels due to clearance and temperature rise considerations, resulting in reduced effectiveness in reducing inductance compared to lower power levels. Therefore, for a more universal inductance

reduction method, further research on PCB layout structures that are not limited by power level should be explored.

**Author Contributions:** Conceptualization, S.-S.M. and C.-H.E.; methodology, S.-S.M. and C.-H.E.; software, S.-S.M. and C.-H.E.; validation, S.-S.M. and C.-H.E.; formal analysis, S.-S.M. and C.-H.E.; investigation, S.-S.M. and C.-H.E.; resources, S.-S.M. and Y.-S.J.; data curation, S.-S.M. and Y.-S.J.; writing—original draft preparation, S.-S.M.; writing—review and editing, S.-S.M.; visualization, S.-S.M. and Y.-S.J.; supervision, R.-Y.K.; project administration, R.-Y.K.; funding acquisition, R.-Y.K. All authors have read and agreed to the published version of the manuscript.

**Funding:** This work was supported by the Korea Institute of Energy Technology Evaluation and Planning (KETEP) grant funded by the Korean government (MOTIE) (20225500000090, Advanced Control & Protection Platform for Multi-terminal MVDC System and Engineering Design Protocol).

**Data Availability Statement:** MDPI Research Data Policies.

**Conflicts of Interest:** The authors declare no conflict of interest.

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
