# Peer review of "Three-Dimensional Lattice Structure to Reduce Parasitic Inductance for WBG Power Semiconductor-Based Converters"

_electronics, doi:10.3390/electronics12081779_

Round 1

Reviewer 1 Report

This is a well written manuscript.  The research address an important problem related to magnetic flus cancellation.  This has been a problem which has not been well understood.  The 3D structure proposed is a possible solution.  The authors should discuss as to whether such a structure may be realized with possible processing limitations.

Reviewer 2 Report

This paper presents a three-dimensional lattice structure that can reduce parasitic inductance through horizontal and vertical magnetic flux cancellations, along with a mathematical conceptualization and layout of a printed circuit board.

It is a very good paper, well written and well explained. My only suggestion targets the introduction. The related work review should be improved, more titles should be reported. Also, a better description of the work proposed in this paper should be given in the final paragraph.

The mathematical model in section 2 is well derived.

The proposed structure and design procedure is good, the description is sufficient and clear.

The results are credible and validate the claims in the abstract and introduction.

My resolution is that the paper is acceptable as is. However, the indicated additions to the introduction would be an improvement.

Reviewer 3 Report

Authors have done well to show a simple concept to implement reduced inductor parasitics in PCB design. The methodology is sound and the concept is clear. I have two questions for the authors:

1. Is there a downside to this design? I suspect R and C parasitics could be problematic if the design method is taken to extremes.

2. If not limited by PCB standards, what would an optimum design look like?

Reviewer 4 Report

The paper deals with parasitic inductance reduction methods in semiconductor-based power converters.

Remarks:

- In the introduction of the paper, only three similar methods to the proposed technique were shown for the problem. These papers support the paper's content but are over ten years old. Please, insert some newer citations to support the novelty of the proposed solution.

- in 2.2 please show the solved maxwell equations, is it only a steady state magnetic field simulation?

- in figure 4, the authors wrote they are showing the measured values. This is a simulation result, not a measurement. Please change the description because I think it is misleading.  Simulation codes can give misleading results or unexpected differences from a measured values.

- in figure 13 it would be better to describe the name of the applied method, than the given reference, it is not obvious that given reference contains only one approach.

Formatting issues:

 - the applied font size is differs in the Introduction and the second section, please follow the template more strictly. These formatting issues devaluate the contant of the paper, however it seems a good work.

Reviewer 5 Report

This paper proposes a three-dimensional lattice structure that can eliminate parasitic inductance in printed circuit boards (PCBs) through horizontal and vertical magnetic flux. A design example fully illustrates the selection of line segment parameters and the application of the proposed structure in actual PCB layout. The article has clear logic and correct format, and can be published after minor modifications. The comments are as follows:

1. Volume number and page information of some references are incomplete.

2. Please explain why reference 15 is cited as the basis for the conventional structure.

3. It is suggested to elaborate the application of wide band gap(WBG) materials in the introduction.(https://doi.org/10.3390/polym14173590)

4. The authors could insert more numerical date into the Abstract for enhancement of the manuscript.

5. Advantages and disadvantages of 3-D lattice structure should be compared in the conclusions, which are described in literature. Authors are suggested to described some future plans in conclusions.
